# Mechanical and Durability Properties of Portland Limestone Cement (PLC) Incorporated with Nano Calcium Carbonate (CaCO_3_)

**DOI:** 10.3390/ma14040905

**Published:** 2021-02-14

**Authors:** Lochana Poudyal, Kushal Adhikari, Moon Won

**Affiliations:** Department of Civil, Environmental and Construction Engineering, Texas Tech University, Lubbock, TX 79416, USA; kushal.adhikari@ttu.edu

**Keywords:** nanotechnology, nano CaCO_3_, Portland limestone cement, workability, mechanical properties, performance, durability

## Abstract

Despite lower environmental impacts, the use of Portland Limestone Cement (PLC) concrete has been limited due to its reduced later age strength and compromised durability properties. This research evaluates the effects of nano calcium carbonate (CaCO_3_) on the performance of PLC concrete. The study follows a series of experiments on the fresh, hardened, and durability properties of PLC concrete with different replacement rates of nano CaCO_3_. Incorporation of 1% nano CaCO_3_ into PLC concrete provided the optimal performance, where the 56 days compressive strength was increased by approximately 7%, and the permeability was reduced by approximately 13% as compared to Ordinary Portland Cement (OPC) concrete. Further, improvements were observed in other durability aspects such as Alkali-Silica Reaction (ASR) and scaling resistance. Additionally, nano CaCO_3_ has the potential to be produced within the cement plant while utilizing the CO_2_ emissions from the cement industries. The integration of nanotechnology in PLC concrete thus will help produce a more environment-friendly concrete with enhanced performance. More in-depth study on commercial production of nano CaCO_3_ thus has the potential to offer a new generation cement—sustainable, economical, and durable cement—leading towards green infrastructure and global environmental sustainability.

## 1. Introduction

Approximately 4.1 billion metric tons of cement are currently produced globally every year, accounting for about eight to ten percent of global anthropogenic carbon-dioxide (CO_2_) emissions [1]. Cement production is expected to increase in the future, adding more CO_2_ into the atmosphere. Sustainable approaches are, therefore, required to effectively control the environmental impact of cement production. A recent report published by Chatham House [2] estimates that the clinker factor needs to be reduced by 60% of the present value by 2050 in order to meet the expected environmental liabilities for cement production. 

Portland Limestone Cement (PLC) has often been used as an environment-friendly alternative to Ordinary Portland Cement (OPC) [3]. PLC is produced by inter-grinding Portland cement clinkers with limestones [4], replacing a certain percentage of the energy intensive cement clinkers with raw limestone. Limestone powder in PLC contributes to improved early age strength by densifying the microstructure due to compact particle distribution and reduces embodied CO_2_ of the cement [4].

European countries were the early users of PLC. The European Standard EN 197 allows two types of Portland limestone cements: Type II/A-L containing 6–20% and Type II/B-L containing 21–35% limestone powder [5]. Likewise, in the United States, the first commercial production of PLC was under performance-based specification ASTM C1157 in 2005 [6]. In 2007, ASTM C150 [7] allowed the replacement of OPC with a maximum of 5% by mass [4]. More recently in 2012, ASTM C595 [8] allowed for the replacement up to 15% for PLC [7]. However, studies show that later age strength and durability in concrete are compromised when cement is replaced by limestone powder in excess of 10% [9,10,11].

To compensate for these limitations, use of other Supplementary Cementitious Materials (SCMs) has been a common practice. One of the commonly used SCM is fly ash, and several studies have concluded that addition of fly ash helps improve concrete properties [4,12]. However, the availability of fly ash will be limited in the future as many sources of fly ash are being shut down [13,14]. Additionally, the availability of fly ash is largely regional thus adding extra costs for the industries for transportation. In recent years, use of nanomaterials has emerged as a promising way to improve the performance of concrete by changing the concrete matrix at nano levels [15,16]. Nano particles have a high surface area to volume ratio; thus, they are more reactive than micro size materials, providing better mechanical performance [17,18].

This paper evaluates the use of nano calcium carbonate (CaCO_3_) as partial replacements (replacement rates of 1%, 2%, and 3%) of PLC in concretes and identifies changes in fresh and hardened concrete properties including durability. Nano CaCO_3_, as compared to several other nano materials such as nano TiO_2_, nano SiO_2_, nano Al_2_O_3_, nano Fe_3_O_4_, nano ZrO_2_, carbon nanotubes, and carbon nanofibers, is economical and easily available [19,20,21,22]. Further, previous studies have shown that nano CaCO_3_ can provide comparable benefits to other higher-cost nanomaterials [23,24]. Additionally, nano CaCO_3_ can be produced within the cement plant utilizing the waste CO_2_ using an integrated approach for green and economical cement production [25,26], thus is an environment-friendly technology. This approach will help mitigate the adverse impacts of cement plants on the environment while also providing extra revenue for the cement plant owners. This paper therefore mainly examines if the use of nano CaCO_3_ can mitigate the shortcomings of PLC concrete.

To the authors’ knowledge, the performance of PLC incorporated with nano CaCO_3_, has not yet been fully explored. Further, minimal efforts have been done to evaluate the durability aspects of using nano CaCO_3_ [16]. This research, therefore, carries significance in providing comprehensive testing results for PLC with nano CaCO_3_, including durability properties and microstructure analysis using SEM (Scanning Electron Microscope) images. While advancing the current knowledge in the field, the findings from this research provide research directions for the future to achieve the sustainability goals in the construction industries.

## 2. Materials and Evaluation Methods

### 2.1. Materials

Type I/II-OPC with Blaine fineness of 390 m^2^/kg and Type IL-PLC with Blaine fineness of 450 m^2^/kg were used as binders. Commercially available Type I/II-OPC was used while Type IL PLC was obtained from a local source and was produced by inter-grinding Type I cement clinker with 15% of limestone by weight in the cement plant. Table 1 shows the chemical compositions of both OPC (Lafarge, Ravena, NY, USA) and PLC (Capitol Aggregates Inc., Austin, TX, USA) used in this study. Limestone with a nominal maximum aggregate size of 19 mm (0.75 in) and siliceous sand with a fineness modulus of 2.6 were used as coarse and fine aggregates, respectively. The gradation of coarse aggregate is shown in Figure 1. White precipitated nano CaCO_3_ (98% pure) with an average diameter of 40 nm and a specific surface area greater than 40 m^2^/g was used. On average, the size of nano CaCO_3_ is 2.5 times smaller, and the specific surface area is twice as large as those of silica fume. Figure 2 shows the microstructure of nano CaCO_3_ under SEM.

### 2.2. Evaluation Methods

The properties evaluated in this study include: (a) slump and setting of fresh concrete, (b) compressive strength and modulus of elasticity of hardened concrete, and (c) concrete durability properties—permeability, alkali silica reaction (ASR), and scaling resistance. In addition, the microstructure was investigated using SEM images and EDS analysis. Concrete samples were prepared as per ASTM guidelines for both OPC and PLC and consisted of samples without nano CaCO_3_ (control sample) and samples with nano CaCO_3_ at varying replacement rates of 1%, 2%, and 3%.

To evaluate the effects of uniform distribution of nano CaCO_3_ on concrete properties, two different concrete mixing methods were evaluated. In one method, CaCO_3_ was introduced to the mixer right after cement addition, which is called regular mixing in this paper. In the other method, nano CaCO_3_ and cement were placed in an electrically driven mechanical mixer and blended under high speed for three to four minutes; this is called modified mixing in this paper. This modified mixing is considered to disperse nano CaCO_3_ more effectively, reducing the agglomeration of nano materials as observed in a concrete made by a regular mixing (Figure 3). This agglomeration or poor dispersion of nano CaCO_3_ will have diminishing effects of nano CaCO_3_.

The mixture proportions for 0.76 m^3^ (1 cy) of concrete are given in Table 2. Water to binder ratio was fixed at 0.47, with a total binder content of 295 kg per 0.76 m^3^ (650 lbs per 1 cy) of concrete. Despite the common practice of using chemical admixtures in concrete production, no chemical admixtures were used in this experiment to limit the interference of other chemicals on the effects of using nano CaCO_3_. A study by Shaikh et al. [27] shows that addition of different superplasticizers varies the strength of nanomaterials.

Slump and setting tests were conducted as per ASTM C143 [28] and ASTM C403 [29], respectively.

For compressive strength and elastic modulus, cylindrical concrete specimens of 100 × 200 mm^2^ (4 × 8 in^2^) were tested at 3, 7, 28, and 56 days. The specimens were fabricated and moist cured in accordance with ASTM C192 [30]. Rapid Chloride Penetration Test (RCPT) was conducted in accordance with ASTM C1202 [31] using the Giatec Scientific test setup (GIATEC, Ottawa, ON, Canada), where the charges passed through the specimen were automatically calculated at the end of six hours. For this testing, cylindrical specimens 100 × 200 mm^2^ (4 × 8 in^2^) were cured for 56 days, as per ASTM C192 [30]. The cured specimens were then cut to the proper size for testing 100 × 50 mm^2^ (4 × 2 in^2^) as prescribed by ASTM C1202 [31]. An alkali silica reaction (ASR) test was performed on prismatic specimens 25 × 25 × 254 mm^3^ (1 × 1 × 10 in^3^) as per ASTM C1260 [32], where sand with high reactivity from a local source was used. The changes in length of specimen were measured as per ASTM C157 [33] at 5, 10, and 14 days. Scaling resistance was evaluated in accordance with ASTM C672 [34]. The testing was performed on beam specimens 150 × 150 × 610 mm^3^ (6 × 6 × 24 in^3^) with plastic dikes of 19 mm (0.75 in) height for the chloride solution.

SEM image and Energy Dispersive Spectroscopy (EDS) analyses were conducted to investigate the effects of nano CaCO_3_ on the microstructural changes in concrete. The specimens were prepared as per ASTM C305 [35]. A Zeiss Crossbeam 540 FIB-SEM (ZEISS, Oberkochen, Germany) was used for obtaining the SEM images and performing the corresponding EDS analysis.

Test results were analyzed to evaluate the effects of nano CaCO_3_ and its replacement rates on fresh and hardened concrete properties, including mechanical properties and durability of concretes made with OPC and PLC. All the testing results presented in this paper are the averages of three specimens, except for scaling resistance (single specimen).

## 3. Results and Discussions

### 3.1. Concrete Slump

Figure 4 shows the slump for different mixes used in the study. In general, the slump values were observed to be low considering the w/b ratio (0.47) used in this study. In this study, no water reducers were used, which may explain the low slump observed in this study. As illustrated in Figure 4, the slump for PLC was larger than that for OPC. A previous study showed that, compared with OPC concrete, comparable slumps were achieved for PLC with a reduced w/b ratio, which improved the mechanical and durability properties of PLC concrete [36]. However, fineness values are different for both cement types, so a general conclusion on the effect of limestone powder in cement on slump cannot be made.

Figure 4 also illustrates the effects of nano CaCO_3_ on slump. Since the specific surface area of nano CaCO_3_ is about twice as large as that of silica fume, the incorporation of nano CaCO_3_ in concrete could adversely affect the workability of concrete, which can be an issue from a practical standpoint. Slump loss of 13 mm (0.5 in) was observed for both OPC and PLC at replacement rates of 2% and 3%. It appears that large surface areas of nano CaCO_3_ have negative effects on concrete workability, as observed in concretes with silica fume. However, 1% replacement of nano CaCO_3_ did not change the slump for either concretes, indicating that the effect of large surface area of nano CaCO_3_ on workability is minimal at a 1% replacement rate.

### 3.2. Setting

Concrete setting is controlled by the hydration rate, where the set time varies inversely with the hydration rate, i.e., an increase in the hydration rate leads to reduced set times. Figure 5 shows the variations in set times for the mixes evaluated in this study. In general, set times for PLC mixes are smaller than OPC mixes, indicating that hydration rates of PLC mixes are higher than those of OPC mixes. It should, however, be noted that the set times might have been affected due to the differences in fineness of the two cements. PLC concrete has a 17% reduction in initial and a 13% reduction in final set times as compared to OPC concrete. Limestone powder (average diameter: 15 µm) in PLC produces several effects on the mechanism and kinetics of cement hydration such as filler effect, nucleation effect, and dilution effect [37].

Limestone powder fills the microstructure space between hydration products, reducing the set time [38]. Limestone powder also provides nucleation sites for the precipitation of hydration products and thus accelerates the hydration, which is known as a nucleation effect [19]. In research studies, limestone powder was used as an accelerator due to this nucleation effect [39,40]. Replacement of cementitious materials with non-cementitious materials, such as limestone powder in PLC, also exhibits dilution effects, and fewer hydration products are produced. However, this dilution effect also avails the free water, which reacts with cement particles, thereby increasing the hydration rate [37]. Figure 5 shows that the combined outcome of counteracting filler, nucleation, and dilution effects on setting is the reduction in setting times.

Further, a study by Kakali et al. [41] has shown that the addition of limestone powder increases the hydration rate, and this rate is accelerated with the increase in fineness. Thus, the addition of nano CaCO_3_ would serve as a catalyst for hydration rates. The setting tests conducted on concretes with different replacement rates of nano CaCO_3_ shown in Figure 5 illustrate that the effects are more pronounced on final set than initial set. Final set times were reduced significantly for both OPC and PLC concretes with up to 2% replacement rates. Compared with controls, final set times for OPC and PLC concretes with 2% replacement decreased by approximately 17% and 21%, respectively. However, compared with 2% replacement, initial and final set times for both concretes increased at 3% replacement. Also, it was observed that the difference between the initial and final setting times for the PLC concretes varied markedly with increased percentages of nano CaCO_3_. The exact mechanism has not been identified; however, it could be possibly due to the inefficient dispersion of nanomaterials as no special mixing techniques were used in this study. This could lead to agglomeration of nano particles thus increasing the size from nano to micro as shown in Figure 3. The higher the replacement rate of nano CaCO_3_ in concrete, the higher the chances for agglomeration. Also, it appears that the increased rate of nano CaCO_3_ accentuated the dilution effect resulting in a decreased rate of hydration.

### 3.3. Compressive Strength

Strength of concrete mainly depends on the amount of hydration products, porosity of concrete, and packing of the microstructure. Figure 6 shows the compressive strength of OPC and PLC. In concretes without added nano CaCO_3_, PLC concrete has higher strength than OPC concrete up to 7 days, while at 28 days, both the concretes have comparable strengths. Limestone powder acts as an inert filler, creating a dense structure in concrete through improved packing and helps in the early hydration of cement due to a nucleation effect, thus improving early age strength of PLC. However, at 56 days, the strength of PLC concrete is a little lower than OPC concrete. At later ages, this reduction could be attributed to the pronounced dilution effect being more dominant, resulting in lower strength of PLC concrete.

Replacement of cement, especially with 1% nano CaCO_3_, increased compressive strength for both OPC and PLC concretes. The effects of nano CaCO_3_, however, are more pronounced at the early ages for OPC concrete. Addition of nano CaCO_3_ accelerates the rate of C_3_S hydration as explained by Sato et al. [19], where the protective layer surrounding the C_3_S broke earlier during the induction period. This break reduces the induction period, thereby accelerating the C_3_S hydration rate and increasing the early age strength.

Replacement of nano CaCO_3_ at 1% increased 3-day strength of OPC concrete by 52%; however, the strength increase diminishes with time, with only about 7% increase at 56 days. A similar trend was observed for PLC concrete, but at a lower scale, with 6% increase at 3 days and only 2% at 56 days. The large difference in strength increase of concretes with and without nano CaCO_3_ between early and later ages, especially for OPC concrete, might be due to a more prominent nucleation effect of nano CaCO_3_ at the early ages. Figure 6 also shows that replacement rates of nano CaCO_3_ at 2% and 3% resulted in lower strengths than at 1% replacement rate for both concretes and at all ages. It appears that, as far as concrete strength is concerned, the optimum replacement rate of nano CaCO_3_ is 1%. Even though it is not known why additional nano CaCO_3_ beyond 1% resulted in reduced strength, it could be due to the enhanced dilution and filler effect [38].

Additionally, the test results were compared between two different mixing sequences. The samples with modified mix sequence (MOP-1 and MPL-1) had better strength at all ages, with the maximum increment at three days, as compared with regular mix (OP-1 and PL-1) concrete. The findings thus signify the importance of proper dispersion of nano CaCO_3_ in improving concrete strength or other mechanical properties. Other methods such as sonification could also be used for better dispersion of nano materials [42].

### 3.4. Elastic Modulus

Figure 7 shows the elastic modulus for different concrete mixes. Despite loose correlation with corresponding compressive strengths, no significant variations were observed in the elastic modulus among different mixes. The modulus of elasticity mainly depends upon the type of coarse aggregate and its strength. In this testing, the same coarse aggregate type was used for all mixtures. Thus, the slight variations in elastic modulus among different mixes could be attributed to the effects of nano CaCO_3_ on porosity or denseness of hydrated cement paste. The modulus of elasticity was slightly higher for concrete specimens with nano CaCO_3_ and a modified mixing sequence. For instance, elasticity at three days was approximately 6% higher for MOP-1 as compared to OP-1 and about 4% higher for MPL-1 as compared to PL-1.

### 3.5. Permeability

It is well known that concrete durability depends primarily on the permeability of concrete. For the evaluation of concrete permeability, rapid chloride penetration tests (RCPT) were conducted on six different concrete mixes with 1% replacement rate of nano CaCO_3_ at 56 days: OP-0, OP-1, MOP-1, PL-0, PL-1, and MPL-1. This testing measures concrete resistivity, not permeability; however, a correlation exists between concrete resistivity and permeability [43]. Other replacement rates (2% and 3%) of nano CaCO_3_ were not included in this testing since compressive testing results positively indicated an optimum replacement rate of 1%. This decision was also aided by several other studies [37,44].

Figure 8 presents the testing results. Overall, charges passed are in “moderate or high” ranges, per ASTM C 1202 [31]. The testing results can be summarized as follows: chloride permeability was reduced with (1) 1% replacement of cement with nano CaCO_3_, and (2) modifying the mixing sequence, or effective dispersion of nano CaCO_3_. Replacement of cement with 1% nano CaCO_3_ reduced chloride permeability by 28% and 11% for OPC and PLC, respectively. This reduction could be the effect of enhanced pore structures and more hydration products due to a nucleation effect, as shown in the increased compressive strengths of concretes with 1% nano CaCO_3_ at 56 days. Improved packing density, formation of supplementary hydration products such as calcium carboaluminates, and carbosulfoaluminates, compact structures, and refined pores were some of the changes in microstructure that helped to reduce the porosity of concrete with the addition of nano CaCO_3_ [38,45].

Further reduction in chloride permeability for concretes with modified mixing (MOP-1 and MPL-1) signifies the importance of proper dispersion of the nano particles in concrete mixing on achieving the maximum improvements of concrete properties. Proper dispersion effects were also observed in the compressive strength (Figure 5).

Figure 8 also shows higher chloride permeability of PLC concrete compared with OPC concrete. The enhanced dilution effect in PLC could be one reason for higher chloride permeability for PLC. Further, as cited in Bonavetti et al. [45], chloride ions in the solution react with monocarboaluminate in PLC to form chloroaluminates, thus capturing the chloride ions. This process apparently provides a higher transmissivity of charges for PLC, leading to increased chloride permeability.

As PLC concrete showed higher slump when compared to OPC, the w/b ratio can be reduced for PLC, thus alleviating the adverse effects of limestone powder on chloride permeability of PLC concrete. Also, studies have shown that PLC with a lower w/b ratio provided better performance [36]. More research on this topic could help in improving the permeability of PLC concrete with added nano CaCO_3_.

### 3.6. Alkali Silica Reaction

Alkali silica reaction (ASR) in concrete occurs when reactive silica present in the aggregates reacts with hydroxide ions in the pore solution to form expansive silica gel, which could cause micro cracks in the concrete when moisture is absorbed. The expansion of concrete due to ASR depends on several factors such as the alkali content of the cement, the amount, and the reactivity of the silica present in the aggregates, the availability of moisture, and the porosity of the concrete.

Substantial research has been conducted to identify the mechanisms of ASR [46,47]. The hydroxyl ions attack the silanol groups (Si–OH) and the siloxane bonds (Si–O–Si) of the poorly crystalized silica network in the reactive silica. In the presence of less positive calcium ions, the resulting negative charges are balanced by potassium and sodium ions in the pore solution. This solution is precipitated in the form of gels and crystals in micropores and interfacial transition zones (ITZ) and causes expansion.

Figure 9 presents the testing results from ASTM C1260 [32]. Extremely high expansion was observed for the control specimen. The fine aggregate selected in this evaluation has been known for excessive ASR potential. Even though the replacement of cement with 1% nano CaCO_3_ did not reduce the expansion below an acceptable limit of 0.1%, it reduced the expansion substantially, especially for OPC mixes. The expansion of MOP-1 concrete is about half of the control (OP-0). The exact mechanism of this reduction in expansion for 1% replacement of cement with nano CaCO_3_ is not known. The alkalinity of the soak solution in ASTM C1260 [32] is quite high. Additionally, studies have indicated conflicting effects of calcium carbonates on the alkalinity of the pore solution [48]. Thus, the reduction in expansion could possibly be due to the improved permeability of the concrete with nano CaCO_3_.

A similar effect was observed for PLC concrete with nano CaCO_3_, but on a lower scale. The expansion for MPL-1 was reduced by approximately 18% as compared to PL-0. This reduced effectiveness of nano CaCO_3_ on PLC concrete could be due to the higher permeability of PLC as compared to OPC concrete. On the contrary, the ASR expansion for PL-0 is approximately 22% lower as compared to OP-0, despite its higher chloride permeability.

### 3.7. Scaling Resistance

Scaling is the removal of a concrete surface in repeated freezing and thawing environments, and mainly occurs in concrete structures where salt is used as a de-icing agent. There are four theories involved in scaling of concrete: osmotic pressure theory, hydraulic pressure theory, critical saturation degree theory, and water absorption theory [49,50,51,52]. Among these, the osmotic pressure theory is the most widely accepted. When salt is used as a de-icing agent, salt solution seeps into the larger pores of concrete, known as capillary pores. These pores get filled with salt solution first, thus the concentration of salt will be high in these pores. These capillary pores are transformed into osmotic pressure cells as the salt concentration increases, and they attract water molecules from the surrounding cement gel. The larger the difference in concentration of salt between the capillary solution and water molecules in the surrounding cement gel, the higher the attraction. This process increases the pressure in capillary pores, which in turn exerts pressure on the surrounding cement paste, causing the paste to come off the surface.

There are a number of factors that affect the scaling resistance of concrete, including porosity or strength, w/b ratio, method of curing, air entrainment, and type of coarse aggregates used [53,54]; of these, air entrainment and porosity are the two major critical factors. Small air pockets formed by air-entrainment inside the concrete help to act as pressure relief points by providing paths for water during the freezing period. Reduced permeability in concrete will help control the penetration of salt solution into the concrete, minimizing the pressure exerted by solutions inside the concrete pores. Research studies also indicated that concrete prepared with w/b < 0.3 did not require entrained air even in the presence of deicing salt [55], indicating the porosity or permeability of concrete is one of the critical factors determining concrete’s potential for scaling.

Figure 10 illustrates the mass loss obtained after every five cycles of freezing and thawing; the figure shows a general trend of increasing rate of mass loss in the early cycles, followed by steady and decreased mass loss for all concretes. OPC concrete shows the greatest loss of mass, and PLC with 1% nano CaCO_3_ experienced the least mass loss. In this study, concrete did not contain any entrained air. It is interesting that quite comparable mass losses were observed for OPC with 1% nano CaCO_3_ and PLC with no nano CaCO_3_. However, PLC with 1% nano CaCO_3_ significantly reduces the mass loss of concrete due to scaling.

Table 3 illustrates the visual rating of concrete scaling exposed to the calcium chloride solution in freeze and thaw cycles. The visual rating number at the end of 50 cycles for OP-0 concrete was 5 and that for MOP-1 concrete was 4. For PLC concrete, the rating number for PL-0 was 5 and MPL-1 was 2. These results show that 1% nano CaCO_3_ improved scaling resistance for both concretes, even though MOP-1 and PL-0 concretes show a similar trend when it comes to the mass of concrete scaled off (Figure 10), the ratings in Table 3 show that the PL-0 concrete had less severe scaling than MOP-1 concrete.

### 3.8. Microstructure of Cement Paste as Affected by Nano Calcium Carbonates

Nano CaCO_3_ could help modify the microstructure through a series of supplementary reactions.

In OPC concrete, reactions between tri-calcium aluminate (C_3_A) and gypsum form ettringite or calcium sulfoaluminates, and as the gypsum is depleted, calcium sulfoaluminates convert to monosulfoaluminates [36]. However, this conversion to monosulfoaluminates is modified with the addition of nano calcium carbonates in Portland cement. Carbonates present in nano CaCO_3_ react with C_3_A in the concrete to form calcium carboaluminates [29,36]. This consumption of C_3_A in the presence of CaCO_3_ retards or stops the conversion of ettringite to monosulfoaluminates. Moreover, study shows that the addition of CaCO_3_ replaces the sulfate ions in calcium sulfoaluminates and monosulfoaluminates by carbonate ions, further stabilizing the ettringites or calcium sulfoaluminates [36].

Laboratory testing revealed improvements in concrete properties when nano CaCO_3_ is incorporated in the concrete as a partial replacement of cement. Efforts were made to identify modifications in the microstructure responsible for the improvements in the concrete properties, and SEM images of the microstructure of cement paste at three days and 60 days were analyzed. Figure 11a,c,e,g illustrates SEM images at three days of hydration for OP-0, MOP-1, PL-0, and MPL-1, respectively, while Figure 11b,d,f,h shows those at 60 days of hydration for OP-0, MOP-1, PL-0, and MPL-1, respectively. All the samples at three days of hydration show a loose structure with ettringite crystals (as represented by needle-like structures) and unhydrated products (as indicated by the fluffy particles). At 60 days hydration, the images show a compact structure with the amount of embedded hydration products in CSH gel, and in the order of MOP-1 followed by MPL-1, PL-0, and OP-0. Further, the EDS analysis on SEM images (not presented here) at three days of hydration confirms that the needle-like structures were ettringites. Likewise, EDS analysis of the needle like formation observed in SEM images at 60 days revealed that they were mixtures of ettringite and CH crystals for OP-0 and for PL-0, MPL-1, and OP-1, they were mixtures of carboaluminates and ettringite.

For OPC, at three days (Figure 11a,c), the ettringite crystals for MOP-1 are much larger as compared to OP-0, indicating a rapid increase in hydration rate at the early ages that might have resulted in higher early compressive strength and reduced final set time. Further, at 60 days of hydration (Figure 11b,d), the SEM image for MOP-1 shows a more compact structure as compared to OP-0, contributing to a denser structure, thus improving the permeability and later age strength of the concrete. This phenomenon could also explain reduced chloride permeability, reduced ASR expansion, and improved scaling resistance of concrete with nano CaCO_3_.

Similarly, for PLC, at 3 days (Figure 11e,g), the SEM image for PL-0 has more unhydrated products (as indicated by white fluffy structures) as compared to MPL-1. Also, the ettringite crystals are longer in MPL-1, indicating an increased hydration rate, thus resulting in a reduced final set time. Additionally, for 60 days (Figure 11f,h), the SEM images for both samples show compact structure but with more hydration products embedded in CSH for MPL-1, thereby improving the later age strength and permeability of PLC concrete.

## 4. Conclusions

To make Portland cement environment-friendly and economical, limestone powder is incorporated during the manufacture of cement. This cement type, called PLC (Portland Limestone Cement), has environmental benefits due to the reduced clinker production required. However, the use of PLC has been limited primarily due to its reduced later age strength. The use of CaCO_3_ in PLC concrete improved the strength and performance of concrete thus potentially encouraging wider use of PLC in the construction industry. In this study, three different replacement rates (1%, 2%, and 3%) of nano CaCO_3_ in PLC were investigated where 1% replacement provided the optimal performance, as summarized below.

The workability of PLC concrete was not adversely affected by 1% replacement of cement with nano CaCO_3_. However, with larger replacements of 2% and 3%, concrete workability decreased. The setting time for the concrete, especially the final set, was affected by the incorporation of nano CaCO_3_. The final setting was accelerated for PLC concretes with up to 2% nano CaCO_3_ replacements; however, at 3% replacement, the trend was reversed with longer set times.A comparable 56-day compressive strength was observed for PLC with 1% nano CaCO_3_. The strength, however, exceeded the strength for OPC by 7% when a modified mixing sequence was used.Permeability at 56-days and ASR expansion at 14-days were reduced by approximately 33% and 20%, respectively with the replacement of 1% nano CaCO_3_ in PLC.SEM images showed that nano CaCO_3_ in both OPC and PLC concretes helps in densifying the pore structure and in increasing the amount of hydration products, thus improving the strength and durability properties of both concretes.

Also, the effect of dispersion of nano CaCO_3_ was investigated by modifying the way nano CaCO_3_ was introduced. Compared with adding nano CaCO_3_ to cement during concrete mixing, pre-mixing of nano CaCO_3_ with cement prior to concrete mixing improved both the mechanical properties and the durability of both OPC and PLC concretes.

## 5. Limitations and Recommendations

In this study chemical admixtures were not used. The effect of nano CaCO_3_ in the presence of chemical admixtures was not investigated. Further study is recommended to evaluate the effect of nano CaCO_3_ in combination with different types of admixtures.

Inefficient dispersion of nanomaterials in the concrete mix has been identified as one of the major challenges and process costs for the industries. As observed in this study, pronounced effects on performance of concrete was observed when a modified mixing sequence was adopted. Further research is needed to identify the optimum mixing method to achieve efficient dispersion of the nanomaterials.

## Figures and Tables

**Figure 1 materials-14-00905-f001:**
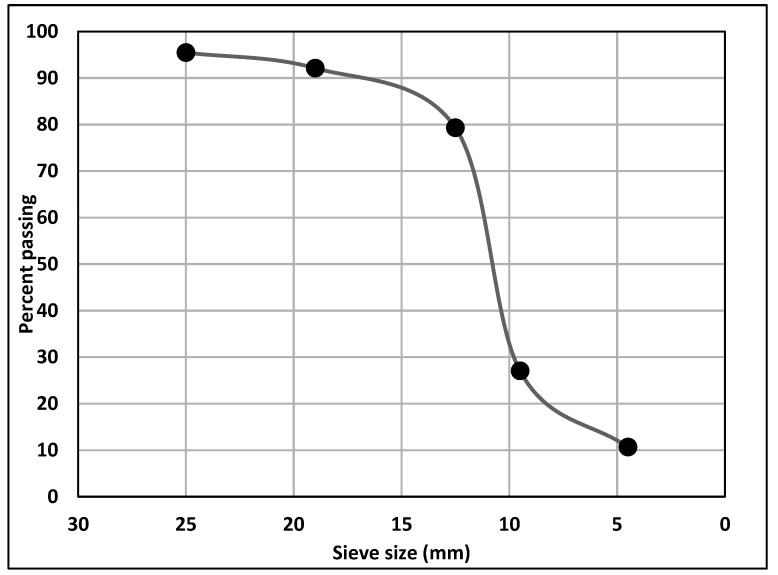
Gradation of coarse aggregate.

**Figure 2 materials-14-00905-f002:**
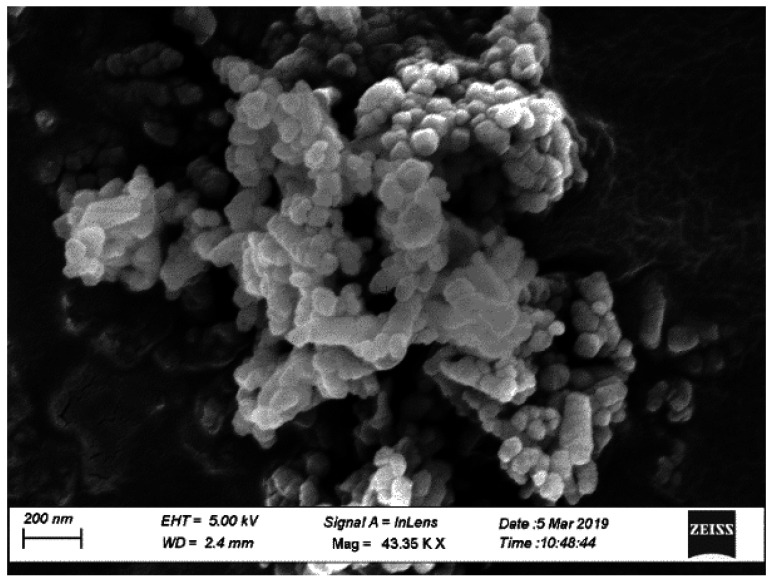
Nano CaCO_3_ under Scanning Electron Microscope (SEM).

**Figure 3 materials-14-00905-f003:**
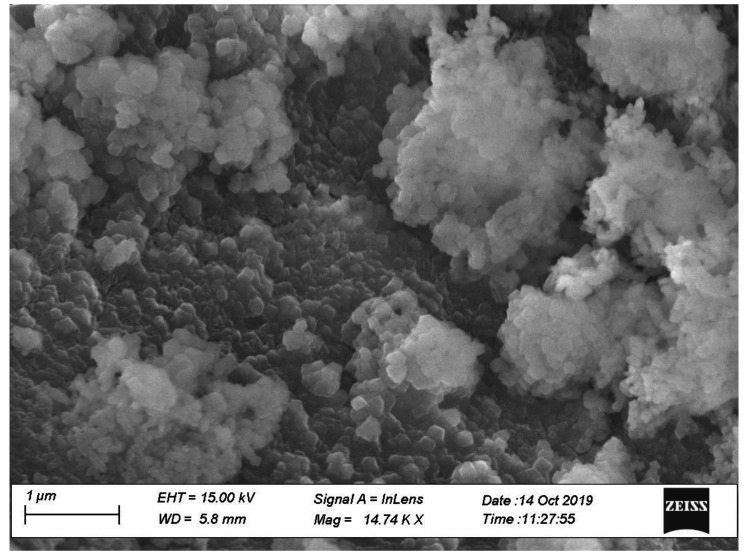
Agglomerated nano CaCO_3_ in the Ordinary Portland Cement (OPC) concrete mix with 1% nano CaCO_3_ (OP-1).

**Figure 4 materials-14-00905-f004:**
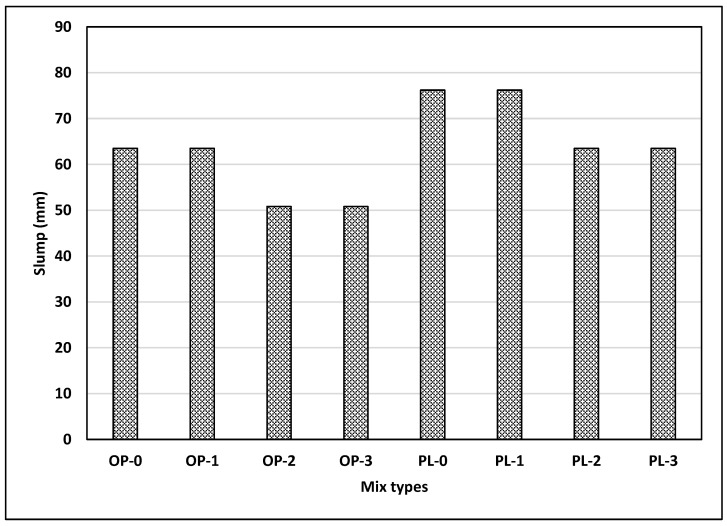
Results of slump test for all mixes.

**Figure 5 materials-14-00905-f005:**
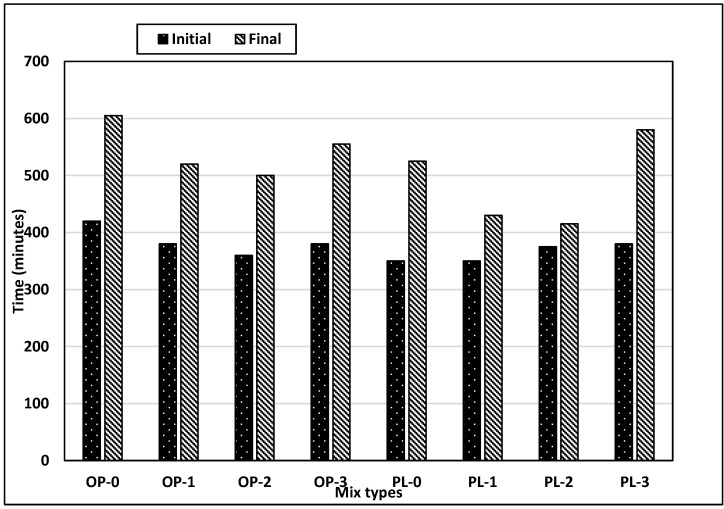
Results of setting test for all mixes.

**Figure 6 materials-14-00905-f006:**
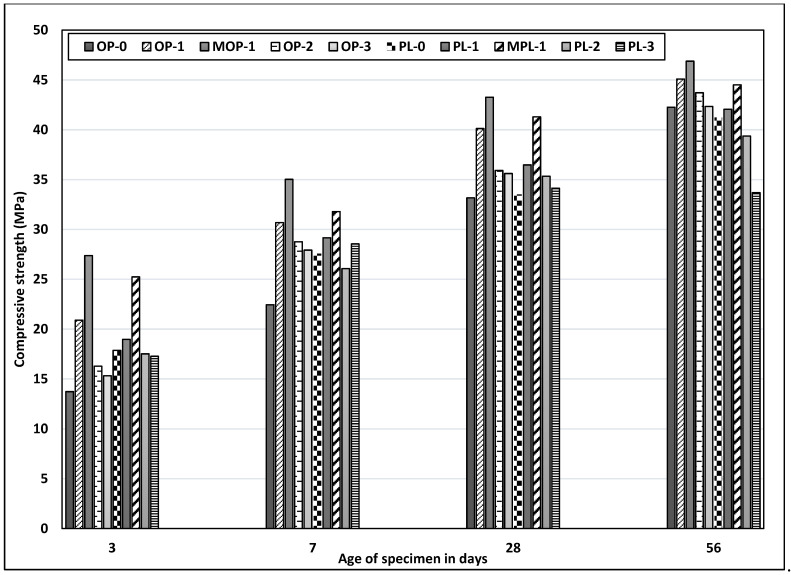
Results of compressive strength test for all mixes.

**Figure 7 materials-14-00905-f007:**
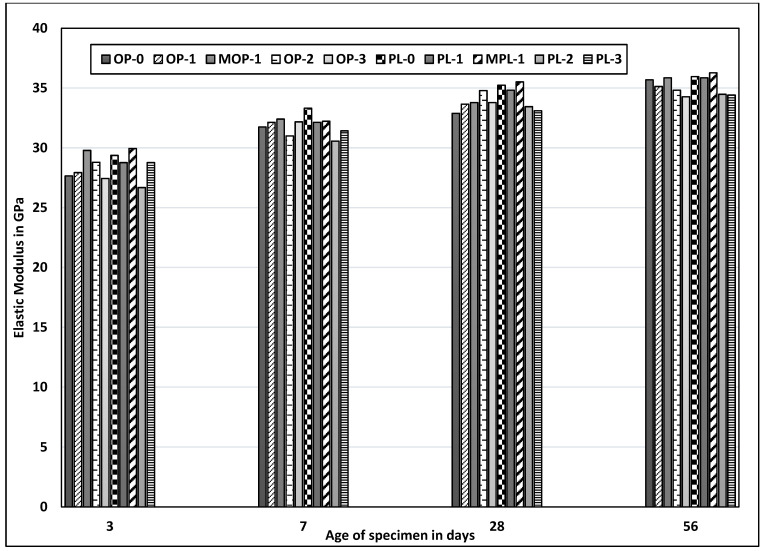
Results of elastic modulus test for all mixes.

**Figure 8 materials-14-00905-f008:**
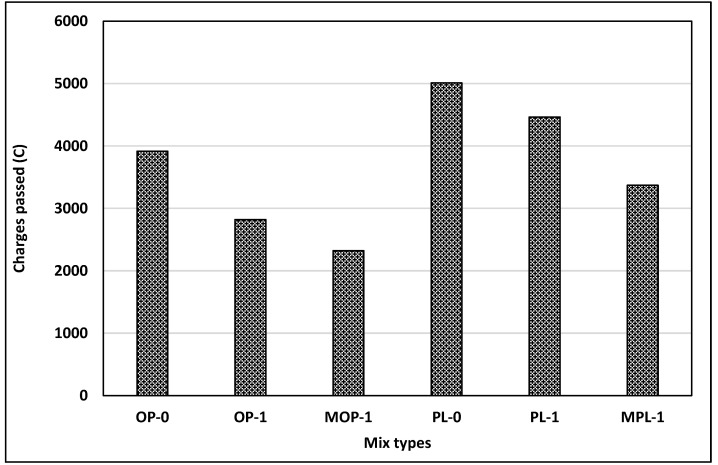
Results of rapid chloride penetration test (RCPT).

**Figure 9 materials-14-00905-f009:**
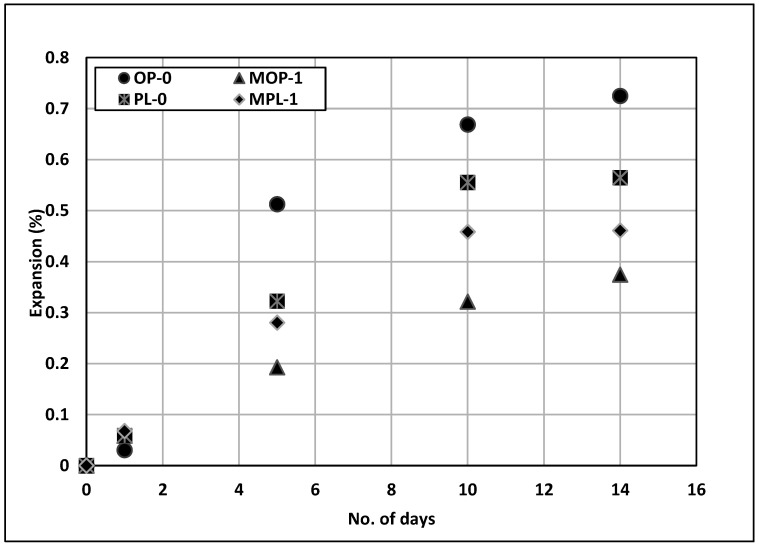
Results of alkali silica reaction (ASR) test.

**Figure 10 materials-14-00905-f010:**
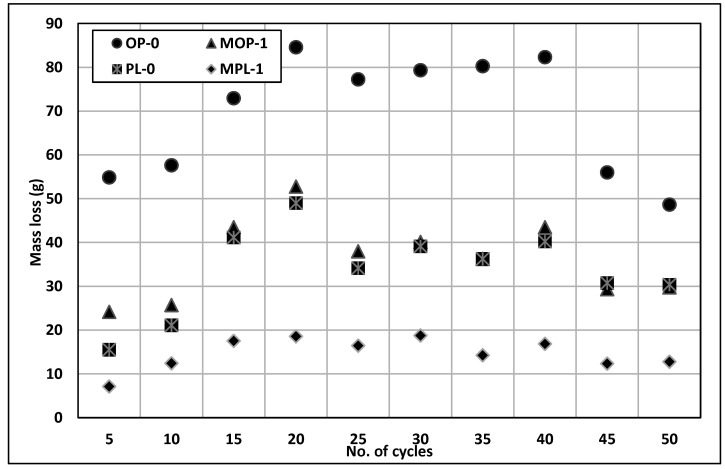
Results of scaling resistance.

**Figure 11 materials-14-00905-f011:**
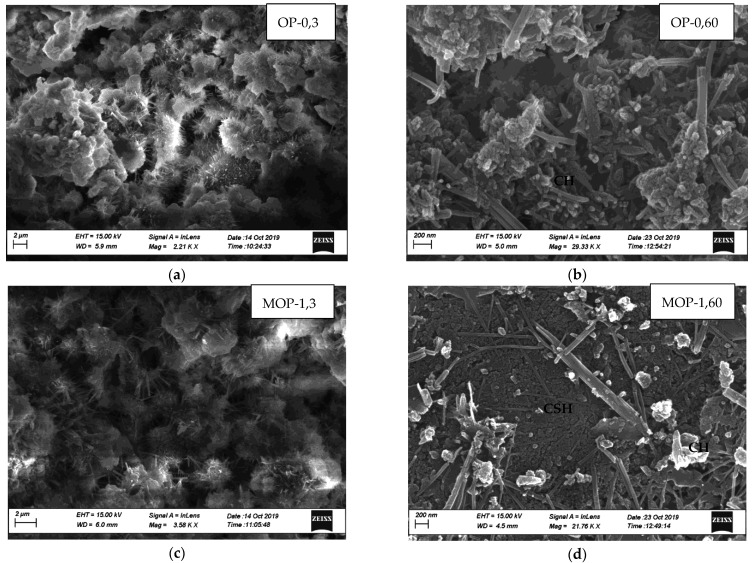
(**a**) Microstructure of OPC at 3 days, (**b**) Microstructure of OPC at 60 days, (**c**) Microstructure of OPC replaced with 1% nano CaCO_3_ at 3 days, (**d**) Microstructure of OPC replaced with 1% nano CaCO_3_ at 60 days, (**e**) Microstructure of PLC at 3 days, (**f**) Microstructure of PLC at 60 days, (**g**) Microstructure of PLC replaced with 1% nano CaCO_3_ at 3 days, (**h**) Microstructure of PLC replaced with 1% nano CaCO_3_ at 60 days.

**Table 1 materials-14-00905-t001:** Chemical composition of Ordinary Portland Cement (OPC) and Portland Limestone Cement (PLC) (% by weight).

Constituents	OPC Type I/II	PLC Type IL
SiO_2_	19.7	20.2
Al_2_O_3_	4.7	5.5
Fe_2_O_3_	3	1.8
CaO	62.1	65
MgO	3.7	1.2
SO_3_	2.9	3.8
Equivalent Alkalis	0.59	0.38
Ignition Loss	2.29	6.1
CO_2_	1.05	0.5

**Table 2 materials-14-00905-t002:** Mix design for 0.76 m^3^ (1 cy) of concrete. Mix designation consists of two parts: cement type (OP for OPC and PL for PLC) followed by a number which represents percentages of nano CaCO3 replaced: w/b ratio = 0.47 for all mixes, all units are in kg.

Mix Designation	Description	Nano CaCO_3_ (kg)	Water (kg)	Cement (kg)	Coarse Aggregates (kg)	Fine Aggregates (kg)
OP-0	OPC with 0% nano CaCO_3_	0	138	295	784	525
OP-1	OPC with 1% nano CaCO_3_	3	138	292	784	525
OP-2	OPC with 2% nano CaCO_3_	6	138	289	784	525
OP-3	OPC with 3% nano CaCO_3_	9	138	286	784	525
MOP-1	OPC with 1% nano CaCO_3_ with modified mix sequence	3	138	292	784	525
PL-0	PLC with 0% nano CaCO_3_	0	138	295	784	525
PL-1	PLC with 1% nano CaCO_3_	3	138	292	784	525
PL-2	PLC with 2% nano CaCO_3_	6	138	289	784	525
PL-3	PLC with 3% nano CaCO_3_	9	138	286	784	525
MPL-1	PLC with 1% nano CaCO_3_ with modified mix sequence	3	138	292	784	525

**Table 3 materials-14-00905-t003:** Rating number for different samples from the scaling resistance test.

Mixes	No. of Cycles				
	5	10	20	25	50
OP-0	1	2	3	4	5
MOP-1	0	1	2	3	4
PL-0	0	1	2	3	3
MPL-1	0	1	1	1	2

## Data Availability

Data is contained within the article.

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
