# Peer review of "Mechanical and Durability Properties of Portland Limestone Cement (PLC) Incorporated with Nano Calcium Carbonate (CaCO3)"

_materials, 2021, doi:10.3390/ma14040905_

Round 1

Reviewer 1 Report

This study proposes a modified PLC by incorporating nano CaCO3 in it for improving the mechanical properties and durability of PLC. A wide range of tests were conducted by the authors. The proposed improving method is interesting, valuable, and seems to me to be potentially promising. I therefore recommend this paper for acceptance, before the authors could respond to the following comments:

(1) The water-to-binder ratio was not small (= 0.47), but the slump values of the OPC and PLC concretes were systematically low. Please explain this result. Besides, the reviewer wonders why the authors didn't use certain superplasticizers. This addition is of practical significance;

(2) The difference between the initial and final setting times for the PLC concretes varied markedly. For example, a 50-minute difference was exhibited for PL-2, but it increased to 200 minutes for PL-3. Please explain this observation. Generic explanations (filler, nucleation, and dilution effects) seem not adequate;

(3) The compressive strength results had the same problem. When it comes down to it, the reason why the subtle variation of nano CaCO3 dosage caused so pronounced different results is vague;

(4) The enhancement of the 56-day compressive strength by adding nano CaCO3 in PLC was virtually insignificant. Thus the barrier of PLC, the reduced later age strength, is not overcome by using nano CaCO3. Please provide comments.

Author Response

Response to Reviewer 1 Comments

Dear Editor and Reviewers,

Thank you very much for your suggestions on the paper. We found the comments very helpful and we have tried our best efforts to address all the comments while improving the quality of the paper. The revised manuscript contains point-by-point responses to the reviewer's comment provided. Please find the responses below for all the comments. 

Reviewer #1 Comment

This study proposes a modified PLC by incorporating nano CaCO3 in it for improving the mechanical properties and durability of PLC. A wide range of tests were conducted by the authors. The proposed improving method is interesting, valuable, and seems to me to be potentially promising. I therefore recommend this paper for acceptance, before the authors could respond to the following comments:

Response: Thank you for your feedback.

Point 1: The water-to-binder ratio was not small (= 0.47), but the slump values of the OPC and PLC concretes were systematically low. Please explain this result. Besides, the reviewer wonders why the authors didn't use certain superplasticizers. This addition is of practical significance.

Response 1: We agree with the reviewer that the slump appears to be low for the given water-to-binder ratio of 0.47. In this study, no water reducers were used, which may explain the low slump observed in this study. Please refer to section 3.1 on page 7. The following verbiage has been added to the manuscript.

Figure 4 shows the slump for different mixes used in the study. In general, the slump values were observed to be low considering the w/b ratio (0.47) used in this study. In this study, no water reducers were used, which may explain the low slump observed in this study.

Also, as mentioned by the reviewer, the use of superplasticizers has been a common practice with practical significance in construction. Yet, we did not use any superplasticizer in our study as our main objective of the paper was to evaluate the sole effects of nano CaCO3. Thus, our goal was to avoid any extra variables or interference on the effects of nano as it has been shown that use of different superplasticizers varies the strength of nanomaterials [47]. Please refer to verbiage added on page 5.

Despite the common practice of using chemical admixtures in concrete production, no chemical admixtures were used in this experiment to limit the interference of other chemicals on the effects of using nano CaCO3. A study by Shaikh & Supit [47] shows that addition of different superplasticizers varies the strength of nanomaterials.

  1. Shaikh, F. U. A., & Supit, S. W. M. , "Effects of superplasticizer types and mixing methods of nanoparticles on compressive strengths of cement pastes.," Journal of materials in civil engineering,, Vols. 28(2), 06015008., 2016.

Point 2: The difference between the initial and final setting times for the PLC concretes varied markedly. For example, a 50-minute difference was exhibited for PL-2, but it increased to 200 minutes for PL-3. Please explain this observation. Generic explanations (filler, nucleation, and dilution effects) seem not adequate.

Response 2: We agree with the reviewers comment as huge variations were observed in the differences between the initial and final setting times for the PLC concrete. Unfortunately, at this point, the authors were not able to identify the exact mechanism for this observation. However, our speculation for higher difference in PL-3 could be due to the inefficient dispersion of nanomaterials. This could lead to agglomeration of nano particles thus increasing the size from nano to micro as shown in Figure 3. And the higher the replacement rate of nano CaCO3 in concrete, the higher the chances for agglomeration. Also, it appears that the increased rate of nano CaCO3 accentuated the dilution effect resulting in decreased rate of hydration. Please refer to Section 3.2 on page 8. The following verbiage has been added to the manuscript.

Also, it was observed that the difference between the initial and final setting times for the PLC concretes varied markedly with increased percentages of nano CaCO3. The exact mechanism has not been identified; however, it could be possibly due to the inefficient dispersion of nanomaterials as no special mixing techniques were used in this study. This could lead to agglomeration of nano particles thus increasing the size from nano to micro as shown in Figure 3. The higher the replacement rate of nano CaCO3 in concrete, the higher the chances for agglomeration. Also, it appears that the increased rate of nano CaCO3 accentuated the dilution effect resulting in decreased rate of hydration.

Point 3: The compressive strength results had the same problem. When it comes down to it, the reason why the subtle variation of nano CaCO3 dosage caused so pronounced different results is vague.

Response 3: If we have understood the comment correctly, we agree with the reviewer that differences in compressive strength were observed with different rates of nano CaCO3. As compared to PL-1, mixes with 2% and 3% nano CaCO3 reduced compressive strength. However, the differences in strength were mostly within 2-5 MPa (except for PL-0 and PL-3 at 56 days). Yet, possible reasonings for the observed differences have been provided in the original manuscript. Please refer to last paragraph in Section 3.3 on page 9. But, when compared with MPL-1, pronounced difference can be observed as mentioned by the reviewer. This difference could be the effects of improved dispersion in MPL-1 as different mix sequence was adopted for MPL-1.

Point 4: The enhancement of the 56-day compressive strength by adding nano CaCO3 in PLC was virtually insignificant. Thus the barrier of PLC, the reduced later age strength, is not overcome by using nano CaCO3. Please provide comments.

Response 4: We agree with the reviewer comments that addition of nano CaCO3 did not result in significant improvement in 56-days compressive strength. The purpose of this study was to evaluate if PLC incorporated with nano CaCO3 could be used as an alternative to OPC. Even though no significant improvements were observed, comparable strength to OPC were observed after the addition of nano CaCO3. Further, with different mixing sequence, the strength exceeded the corresponding strength for OPC by 7% and for PL-0 by 10%. Based on the findings from this study, more studies on proper dispersion of nano CaCO3 have potential to further improve the strength of PLC. It has also been shown that the addition of chemical admixtures helps enhance the performance of concrete with nano materials as it helps disperse the nanomaterials more efficiently [47]. Please refer to newly added Section 5 Limitations and Recommendations on page 18. The added verbiage is as follows:

  1. Limitations and Recommendations

In this study chemical admixtures were not used. The effect of nano CaCO3 in the presence of chemical admixtures was not investigated. Further study is recommended for the evaluation of effect of nano CaCO3 in combination with different type of admixtures.

Inefficient dispersion of nanomaterials in the concrete mix has been identified as one of the major challenges and costly process for the industries. As observed in this study, pronounced effects on performance of concrete was observed when a modified mixing sequence was adopted. Further research is needed to identify optimum mixing method to achieve efficient dispersion of nanomaterials.

Reviewer 2 Report

Reviewer comment

Manuscript ID: materials-1104041

Title: Mechanical and durability properties of Portland Limestone Cement (PLC) incorporated with nano calcium carbonate (CaCO3).

The authors reported the Mechanical and durability properties of Portland Limestone Cement (PLC) incorporated with nano calcium carbonate (CaCO3). After careful reading, I feel that the work is not acceptable for publication in materials in its present form. The following issues should be addressed.

Comments

  • Highlights - nano CaCO3…to nano CaCO3, …..concrete with nano CaCO3.. to concrete with nano CaCO3….., nano CaCO3 produced optimal results….to nano CaCO3 produced optimal results.
  • Reduction in permeability is very less with nano SiO2.
  • Increase in compressive strength (only 7%) not significant.
  • It seems to be there is no predominant increase in the performance of concrete and reduces the workability and increases the setting time.
  • Durability studies may substantiate the positive benefits of nano CaCO3.
  • Cost benefit ratio of adding nano CaCO3 is very less.
  • Addition of mineral admixtures like RHA, SF, and MK may improve the properties.
  • Hydration products like CH and CSH may be shown in SEM image to understand the micro structure of PLC with and without nano CaCO3.
  • Needle like formations in the SEM image at 60 days may be explained.
  • Conclusion - too long, should be concise.

Author Response

Response to Reviewer 2 Comments

Dear Editor and Reviewers,

Thank you very much for your suggestions on the paper. We found the comments very helpful and we have tried our best efforts to address all the comments while improving the quality of the paper. The revised manuscript contains point-by-point responses to the reviewer's comment provided. Please find the responses below for all the comments. 

Reviewer #2 Comment

Manuscript ID: materials-1104041

Title: Mechanical and durability properties of Portland Limestone Cement (PLC) incorporated with nano calcium carbonate (CaCO3).

The authors reported the Mechanical and durability properties of Portland Limestone Cement (PLC) incorporated with nano calcium carbonate (CaCO3). After careful reading, I feel that the work is not acceptable for publication in materials in its present form. The following issues should be addressed.

 Response: Thank you for your feedback.

Comments

Point 1: Highlights - nano CaCO3…to nano CaCO3, …..concrete with nano CaCO3.. to concrete with nano CaCO3….., nano CaCO3 produced optimal results….to nano CaCO3 produced optimal results.

Response 1: Thank you for pointing out the typo. All such typos have been fixed in the revised manuscript. Please refer to Highlights on page 1.

Highlights

Evaluated the performance of PLC concrete incorporated with nano CaCO3.

Nano CaCO3 in PLC concrete improved later age strength and durability properties.

SEM images showed the densified microstructure in PLC concrete with nano CaCO3.

Replacement of PLC by 1% nano CaCO3 produced optimal results.

Integration of nanotechnology in PLC concrete will help produce more environment-friendly concrete with enhanced performance.

Point 2: Reduction in permeability is very less with nano SiO2.

Response 2: We assume that the reviewer meant to say nano CaCO3 when it says nano SiO2. The permeability test was conducted for two different mix sequences. Compared with PLC without nano CaCO3, permeability was reduced by approximately 33% when a modified mixing sequence was used (MPL-1), while the reduction was around 25% with regular mixing sequence (PL-1). This shows that dispersion of nano CaCO3 has significant effects on the performance of concrete. Thus, more studies with improved dispersion have potential to further reduce the permeability. Also, compared to control mix (OP-0), the permeability of PLC with 1% nano CaCO3 was reduced by about 15%. Please refer to newly added Section 5 Limitations and Recommendations.

  1. Limitations and Recommendations

In this study chemical admixtures were not used. The effect of nano CaCO3 in the presence of chemical admixtures was not investigated. Further study is recommended for the evaluation of effect of nano CaCO3 in combination with different type of admixtures.

Inefficient dispersion of nanomaterials in the concrete mix has been identified as one of the major challenges and costly process for the industries. As observed in this study, pronounced effects on performance of concrete was observed when a modified mixing sequence was adopted. Further research is needed to identify optimum mixing method to achieve efficient dispersion of nanomaterials.

Point 3: Increase in compressive strength (only 7%) not significant.

Response 3: We agree with the reviewer comments that addition of nano CaCO3 did not result in significant improvement in 56-days compressive strength. The purpose of this study was to evaluate if PLC incorporated with nano CaCO3 could be used as an alternative to OPC. Though no significant improvements were observed, comparable strength to OPC were observed after the addition of nano CaCO3. Further, with different mixing sequence, the strength exceeded the corresponding strength for OPC by 7% and by 10% when compared to PL-0. Based on the findings from this study, more studies on proper dispersion of nano CaCO3 has potential to further improve the strength of PLC. Additionally, it has been shown that the addition of admixtures help enhance the performance of concrete with nano materials [47]. Please refer to newly added Section 5 Limitations and Recommendations.

  1. Shaikh, F. U. A., & Supit, S. W. M. , "Effects of superplasticizer types and mixing methods of nanoparticles on compressive strengths of cement pastes.," Journal of materials in civil engineering,, Vols. 28(2), 06015008., 2016.

Point 4: It seems to be there is no predominant increase in the performance of concrete and reduces the workability and increases the setting time.

Response 4: The objective of this study was to evaluate the effects of using nano CaCO3 in PLC concrete as very few or no studies have yet been done on use of nano CaCO3 on PLC. Even though improvements observed in the testing of PLC concrete conducted in this study are not significant in certain properties, most of the test results exhibited comparable or even better performance when compared to OPC. Additionally, no chemical admixture and special mixing techniques were used in this study. Thus, future studies focused on enhancing the dispersion of nano CaCO3 on PLC might be able to further improve its performance. Please refer to newly added Section 5 Limitations and Recommendations.

In regard to workability, no significant difference was observed in the slump with addition of 1% nano CaCO3. Reduced slump was, however, observed at higher replacement rates which could be attributed to the negative effects of large surface areas of nano CaCO3. Lastly, reduction in setting time were observed for both the concretes with up to 2% replacement rates.

Point 5: Durability studies may substantiate the positive benefits of nano CaCO3.

Response 5: Same as response in point 4.

The objective of this study was to evaluate the effects of using nano CaCO3 in PLC concrete as very few or no studies have yet been done on use of nano CaCO3 on PLC. Even though improvements observed in the testing of PLC concrete conducted in this study are not significant in certain properties, most of the test results exhibited comparable or even better performance when compared to OPC. Additionally, no chemical admixture and special mixing techniques were used in this study. Thus, future studies focused on enhancing the dispersion of nano CaCO3 on PLC might be able to further improve its performance. Please refer to newly added Section 5 Limitations and Recommendations.

Point 6: Cost benefit ratio of adding nano CaCO3 is very less.

Response 6: Nanotechnology has emerged as a relatively new technology in improving the performance of concrete. Also, compared to other nanomaterials such as nano SiO2, the cost of nano CaCO3 is substantially low, as the cost for nano CaCO3 used in this study is $0.67/lb. Further, this study was mainly aimed at evaluating the performance of concrete and do not look in detail the economic aspect of using nano CaCO3. Future studies evaluating the cost benefit ratio of adding nano CaCO3 will help industries plan for economical alternatives.

Point 7: Addition of mineral admixtures like RHA, SF, and MK may improve the properties.

Response 7: We agree with the reviewer that mineral admixtures like RHA, SF, and MK might improve the properties due to their pozzolanic reactions. However, the use of aforementioned SCMs were not investigated in this study and is beyond the scope of our paper. Further research in this area could help identify further improvements of PLC concrete.

Point 8: Hydration products like CH and CSH may be shown in SEM image to understand the micro structure of PLC with and without nano CaCO3.

Response 8: It has been shown in the images that are applicable. Please refer to Figure 11 on Page 15 and Page 16.

Point 9: Needle like formations in the SEM image at 60 days may be explained.

Response 9: Needle like formation in OP-0 appears to be CH crystal and ettringite whereas for other specimens like PL-0, MPL-1, and OP-1, EDS analysis showed it to be carboaluminates. The following verbiage has been added in the manuscript. Please refer to Page 15.

EDS analysis of the needle like formation observed in SEM images at 60 days revealed that they were the mixtures of ettringite and CH crystals for OP-0 and for PL-0, MPL-1, and OP-1, they were the mixtures of carboaluminates and ettringite.

Point 10: Conclusion - too long, should be concise.

Response 10: The conclusion has been revised. Also, a new section Limitations and Future Recommendations has been added in the revised manuscript. Please refer to Section 4 on page 16 and Section 5 on page 17. The following is the revised conclusion section.

  1. Conclusions

To make Portland cement environment-friendly and economical, limestone powder is incorporated during the manufacture of cement. This cement type, called PLC (Portland Limestone Cement), has environmental benefits due to reduced clinker production required. However, PLC has been limitedly used primarily due to its reduced later age strength. The use of CaCO3 in PLC concrete improved the strength and performance of concrete thus potentially encouraging wider use of PLC in the construction industry. In this study, three different replacement rates (1%, 2%, and 3%) of nano CaCO3 in PLC were investigated where 1% replacement provided the optimal performance, as summarized below.

  1. The workability of PLC concrete was not adversely affected by 1% replacement of cement with nano CaCO3. However, with larger replacements of 2% and 3%, concrete workability decreased. Setting time for concrete, especially final set, was affected by the incorporation of nano CaCO3. Final setting was accelerated for PLC concretes with up to 2% nano CaCO3 replacements; however, at 3% replacement, the trend was reversed with longer set times.  
  2. A comparable 56-day compressive strength was observed for PLC with 1 % nano CaCO3. The strength, however, exceeded the strength for OPC by 7% when modified mixing sequence was used.
  3. Permeability at 56-days and ASR expansion at 14-days were reduced by approximately 33% and 20%, respectively with the replacement of 1% nano CaCO3 in PLC.
  4. SEM images showed that nano CaCO3 in both OPC and PLC concretes helps in densifying the pore structure and in increasing the amount of hydration products, thus improving the strength and durability properties of both concretes.

Also, the effect of dispersion of nano CaCO3 was investigated by modifying the way nano CaCO3 was introduced. Compared with adding nano CaCO3 to cement during concrete mixing, pre-mixing of nano CaCO3 with cement prior to concrete mixing improved both mechanical properties and durability of both OPC and PLC concretes. 

  1. Limitations and Recommendations

In this study chemical admixtures were not used. The effect of nano CaCO3 in the presence of chemical admixtures was not investigated. Further study is recommended for the evaluation of effect of nano CaCO3 in combination with different type of admixtures.

Inefficient dispersion of nanomaterials in the concrete mix has been identified as one of the major challenges and costly process for the industries. As observed in this study, pronounced effects on performance of concrete was observed when a modified mixing sequence was adopted. Further research is needed to identify optimum mixing method to achieve efficient dispersion of nanomaterials.

Round 2

Reviewer 2 Report

Reviewers' comments:

The manuscript can published. The authors have answered the questions.

So that I recommended this manuscript accept for publication in Materials.